# *In-Situ* Sludge Reduction in Membrane-Controlled Anoxic-Oxic-Anoxic Bioreactor: Performance and Mechanism

**DOI:** 10.3390/membranes12070659

**Published:** 2022-06-27

**Authors:** Chengyue Li, Tahir Maqbool, Hongyu Kang, Zhenghua Zhang

**Affiliations:** 1Institute of Environmental Engineering & Nano-Technology, Tsinghua Shenzhen International Graduate School, Tsinghua University, Shenzhen 518055, China; li-cy19@mails.tsinghua.edu.cn (C.L.); tahir.maqbool702@gmail.com (T.M.); khy20@mails.tsinghua.edu.cn (H.K.); 2Guangdong Provincial Engineering Research Centre for Urban Water Recycling and Environmental Safety, Tsinghua Shenzhen International Graduate School, Tsinghua University, Shenzhen 518055, China; 3School of Environment, Tsinghua University, Beijing 100084, China

**Keywords:** membrane bioreactor, sludge reduction, endogenous respiration, denitrification, metabolic pathways

## Abstract

Conventional and advanced biological wastewater treatment systems generate excess sludge, which causes socio-economic and environmental issues. This study investigated the performance of membrane-controlled anoxic-oxic-anoxic (AOA) bioreactors for in-situ sludge reduction compared to the conventional anoxic-oxic-oxic membrane bioreactor (MBR_control_). The membrane units in the AOA bioreactors were operated as anoxic reactors at lower sludge recirculation rates to achieve hydrolysis of extracellular polymeric substances (EPS) and extensive endogenous respiration. Compared to MBR_control_, the AOA bioreactors operated with 90%, and 80% recirculation rates reduced the sludge growth up to 19% and 30%, respectively. Protein-like components were enriched in AOA bioreactors while fulvic-like components were dominant in MBR_control_. The growth of *Dechloromonas* and *Zoogloea* genra was promoted in AOA bioreactors and thus sludge reduction was facilitated. Metagenomics analysis uncovered that AOA bioreactors exhibited higher proportions of key genes encoding enzymes involved in the glycolysis and denitrification processes, which contributed to the utilization of carbon sources and nitrogen consumption and thus sludge reduction.

## 1. Introduction

The activated sludge process is well-adapted for domestic and industrial wastewater treatment [1,2,3]. The extensive biomass generation in such retention systems also remains a tricky problem from a socio-economic and sustainable environment perspective since the excess sludge contains various hazardous substances and the management cost is considerably high [4,5,6]. Therefore, more financial investment and consideration are required for sludge handling and disposal [7]. In recent years, the stringent environmental legislation (GB18918-2002) and the 12th Five-Year Plan of China, which began in 2011, have already put forward new policies on excess sludge disposal and management [8].

Nowadays, researchers are working on multiple end-of-pipe solutions for the cost-effective mitigation of excess sludge [1,8]. In this regard, in-situ sludge reduction in biological wastewater treatment systems is an emerging concept in tackling the issue. In the last two decades, several studies have reported the effectiveness of the oxic-settling anoxic (OSA) process to reduce biomass by minimizing bacterial growth. The additional anoxic/anaerobic unit in the OSA process accelerates particular microbial functions, limiting bacterial growth [9]. Although trials are only found at the lab-scale levels, the OSA (as a side-stream reactor) concept has also been extended to MBR. For this purpose, biomass from the membrane tank is passed through an additional anoxic/anaerobic retention reactor [10].

However, additional installation in existing MBRs would raise technical and economic issues. For example, Wang et al. [11] installed an additional anaerobic side-stream reactor (ASSR) with a working volume of 50 L to the MBR, increasing a total volume up to 75% and achieving a sludge reduction of 21.7%. As such, there is a compelling need for a configuration of MBR, which could promote the in-situ sludge reduction while retaining the efficiency of the system. Furthermore, the underlying mechanisms for sludge reduction are complex and still unclear. In general, researchers have proposed that the cryptic growth, maintenance metabolism, destruction of EPS, hydrolysis of particulate organic matters (POMs), predation, endogenous metabolism and uncoupling metabolism are the common causes for sludge reduction. These phenomena were controlled by regulating the slow-growing bacteria, predators, hydrolytic and fermentative microorganisms [12,13,14,15]. Although much attention has been paid on the effect of different operational parameters on sludge minimization, nutrient removal and the dynamics of the microbial community composition in MBR-SSR systems (SSR: side-stream reactor), few studies elucidate the details from the perspective of metabolic pathways in the sludge reduction process. Recently, Zhao et al. [16] investigated the nitrogen metabolic pathways and key functional genes in the MBR-APSSR system (APSSR: ASSR packed with carriers) by metagenomics analysis and found the correlation between nitrogen consumption and sludge reduction. However, no study discusses the carbon metabolism in such systems. Exploring the carbon and nitrogen metabolic pathways could help the understanding of the sludge reduction mechanism.

Based on the abovementioned knowledge gaps, this study evaluated the membrane-controlled anoxic-oxic-anoxic (AOA) bioreactors that performed in-situ sludge reduction without chemicals addition or a side-stream reactor. The specific objectives of the study are: (i) operating the AOA bioreactors at lower sludge interchange rates with the purpose of severe endogenous respiration conditions in the membrane tank (M-tank), (ii) comparing the impact of different sludge interchange rates in different AOA bioreactors on biomass yield, detailed characterization of dissolved organic matter (DOM), (iii) exploring the mechanisms of sludge reduction by 16S rRNA amplicon sequencing and metagenomics.

## 2. Materials and Method

### 2.1. Experimental Setups

As shown in Figure 1, two membrane-controlled anoxic-oxic-anoxic (AOA) bioreactors with a working volume of 7.4 L each were operated in parallel. A conventional anoxic-oxic-oxic MBR_control_ having a scheme of an anoxic unit (2.0 L) (DO < 0.5 mg/L), an oxic unit (3.2 L) (DO of 1.5–2.5 mg/L), and a membrane tank as an oxic chamber (DO of 1.5–2.5 mg/L) with a volume of 2.2 L was considered as control, and mixed liquor shifting from anoxic to oxic units in MBR_control_ was controlled by gravity flow mixing (Figure 1). In membrane-controlled AOA bioreactors, membrane tanks (2.2 L) were operated in anoxic conditions (DO < 0.5 mg/L) following the typical pre-anoxic (2.0 L) and oxic units (3.2 L). In AOA bioreactors, mixed liquor shifting was controlled using peristaltic pumps from anoxic to oxic to M-tank rather than the gravity flow mixing. Lower sludge recirculation (<100%) from the M-tank to the anoxic unit allowed the higher concentration of the biomass in the M-tank than the respective anoxic and oxic units in AOA bioreactors.

One of the AOA bioreactors was operated with a 90% recirculation rate (AOA_90_) and the other with 80% (AOA_80_) (Figure 1). These recirculation rates were calculated based on permeate flux (5 LMH). AOA_90_ and AOA_80_ were operated with the respective recirculation rates of 90% and 80% for 23 h every day; however, for one hour of 24 h every day, the recirculation rates of these lines were increased to 1800% and 1600% for AOA_90_ and AOA_80_, respectively. In this way, famine/feast conditions were controlled in M-tanks of AOA bioreactors. After 23 h of operation each day, the mixed liquor suspended solid (MLSS) concentration was lower in anoxic and oxic units compared to the M-tanks of AOA bioreactors. The recirculation from the oxic to the anoxic unit was kept the same, i.e., 200%, in all bioreactors (MBR_control_, AOA_90_ and AOA_80_). The same synthetic wastewater was used for all MBRs, and the detailed composition is presented in the Appendix A (Appendix A). The synthetic wastewater containing glucose, ammonium chloride (NH_4_Cl), sodium nitrate (NaNO_3_) and potassium dihydrogen phosphate (KH_2_PO_4_) was used as the major source of carbon (DOC: 399 mg/L), nitrogen (TN: 40 mg/L) and phosphorous (TP: 6 mg/L), respectively. Micro-nutrients (i.e., MgSO_4_ and FeCl_3_) were also added in trace amounts. The pH of reactors was maintained at around 7.0 by adding NaHCO_3_ in the feed. The operation comprised two phases, i.e., acclimatization phase (112 days) and stable phase (90 days), which were determined based on MLSS.

Flat sheet ceramic membrane modules were purchased from Meidensha, Japan, with a nominal pore size of 0.1 µm and an effective area of 0.0425 m^2^, and were used in all bioreactors. The operating parameters such as sludge retention time (SRT), hydraulic retention time (HRT) and flux of AOA bioreactors and MBR_control_ were 80 days, 34 h, and 5 LMH, respectively. The maximum allowable transmembrane pressure (TMP) was designed to be 30 kPa, after which membrane modules were both physically and chemically cleaned.

### 2.2. Extraction and Characterization of EPS and SMP

The supernatant containing soluble microbial product (SMP) was separated from sludge flocs by centrifuging the mixed liquor at 4000 rpm and 4 °C. EPS was extracted by adopting the so-called heating method described in Xiao et al. [17], with some modifications. The flocs were re-suspended in a 0.05% NaCl solution, and heated at 70 °C in a water bath for 30 min, and centrifuged at 4000 rpm to collect the bound EPS. All these EPS were filtered through a 0.45 μm membrane filter and stored at −10 °C for further analytical measurements. The protein and polysaccharide contents in EPS and SMP were characterized using Lowry and phenol/H_2_SO_4_ methods, respectively [18,19].

### 2.3. Analytical Characterization

The pH of mixed liquor was regularly monitored using a desktop pH meter (Thermo, Waltham, MA, USA). Dissolved organic carbon (DOC) and total dissolved nitrogen (TN) were measured using the TOC-V analyzer (Shimadzu, Kyoto, Japan). The nutrients, including ammonium (NH_4_^+^), nitrate (NO_3_^−^) and total phosphate (TP) concentrations, were determined using a colorimetric method as given in the Standard Examination Methods for Drinking Water—Nonmetal Parameters, China (GB/T 5750.5-2006). The mixed liquor suspended solids (MLSS) were measured using APHA standard methods [20].

The significant difference was analyzed with the Origin software using one-way ANOVA test. Principle component analysis (PCA) was also implemented by the Origin software.

### 2.4. PARAFAC Modeling

Fluorescence spectra in the form of a three-dimensional excitation-emission matrix (EEM) were collected from a fluorescence spectrometer (F-7000 Hitachi, Tokyo, Japan). The fluorometer was operated with the following adjusted parameters: excitation (Ex) and emission (Em) wavelength ranges were set to 220–500 nm and 240–550 nm with the incremental step of 5 nm and 1 nm, respectively. The scan speed was 12,000 nm/min, and the slit size (Ex/Em) was fixed at 10 nm. The EEMs of blanks (Milli-Q) were also measured regularly and subtracted from the EEMs of samples. A dataset consisting of 326 EEMs of EPS and SMP from three MBRs was used for PARAFAC modeling. A free toolbox, DOMFlour, was used in Matlab (MathWorks Inc., Natick, Massachusetts, USA), and a tutorial by Stedmon and Bro (2008) was followed. Detailed steps could be found elsewhere [21,22].

### 2.5. 16S rRNA Amplicon Sequencing and Metagenomics Analysis

The sludge samples under steady state at different stages of the experiment, i.e., 0, 45th, and 90th day of operation, were collected from each reactor for microbial analysis. The sludge samples were analyzed by MAGIGENE (Guangzhou, China) for high-throughput sequencing. Deoxyribonucleic acid (DNA) was extracted using the Power Soil DNA Isolation Kit (MOBIO, San Diego, CA, USA). The amplification of the V4-V5 regions of the extracted DNA was obtained using the specific 16S rRNA (515F/907R) as a primer by polymerase chain reaction (PCR). The PCR procedures were conducted using the BioRad S1000 (Bio-Rad Laboratory, Hercules, CA, USA) following the protocol developed by the manufacturer. The purification of the PCR products and the quality control, and construction of sequencing library were obtained by following a previous study of Renetal [23]. The library sequencing was performed using the Illumina Nova Seq 6000 platform and 250 bp paired-end reads were obtained. Finally, a range of detailed DNA sequence analyses were performed in accordance with Ren et al. [23].

The sludge samples in the M-tank of three reactors were collected for metagenomics analysis. In this case, a total of nine samples were obtained for DNA extraction, and the Illumina Miseq 2500 platform was applied for metagenomic sequencing analysis after library preparation. The glucose and nitrogen metabolism pathway maps were determined from the Kyoto Encyclopedia of Genes and Genomes (KEGG) database, and the relative abundances of different functional categories were calculated based on gene annotation results from each database.

## 3. Results and Discussion

### 3.1. Basic Water Quality Parameters

The temporal and average of the long-term system performances of bioreactors in terms of DOC and nutrient removals (TN and TP) are presented in Appendix A. The results are only presented for the stable phase. The results showed effective removal of organic substrates and nutrients (TN and TP) in all three bioreactors. The effluents obtained from these systems had reasonable qualities for discharge [24]. The extensive removal (>99%) of organic substrates could be ascribed to the long designed SRT (i.e., 80 days). Previous studies also achieved refined quality of effluent at such prolonged retention [25,26]. It deduced from the results that the performance of AOA bioreactors remained intact compared to MBR_control_ in removing organics and nutrients. Remarkable TN removal in three bioreactors was achieved with the effluent concentration of 2.43 ± 1.76, 2.19 ± 1.23 and 1.77 ± 1.55 mg/L in MBR_control_, AOA_90_ and AOA_80_, respectively. Although the difference among the TN concentration in effluents was not statistically significant, the AOA bioreactors seemed to be more efficient than MBR_control_. On the other hand, the TP removal was slightly lower in effluents obtained from AOA bioreactors with an average of 0.62 ± 0.46 and 0.65 ± 0.25 mg/L for AOA_90_ and AOA_80_, respectively.

The role of different treatment units (anoxic/oxic/M-tank) in removing TN (NH_4_^+^ and NO_3_^−^) species was also elucidated and presented in Appendix A. In MBR_control_, a gradual decrease in NH_4_^+^ concentration was recorded from anoxic to oxic to M-tank, highlighting the direct role of the oxic condition in removing nutrients. At the same time, NO_3_^−^ was increased from an anoxic to an oxic unit in MBR_control,_ presenting the transformation of NH_4_^+^ (*p* < 0.05). In AOA bioreactors, the anoxic unit showed the highest NH_4_^+^ concentration, 6.71 ± 2.18 and 3.23 ± 2.87 mg/L, and the lowest NH_4_^+^ concentration was found in the oxic unit, 0.47 ± 0.63 and 0.25 ± 0.31 mg/L, in AOA_90_ and AOA_80_, respectively. The extended anoxic condition in AOA bioreactors was found helpful in removing NO_3_^−^ with prolonged denitrification. In general, results proved the efficient working of membrane-controlled AOA bioreactors, which did not halt any general phenomenon in a typical MBR but provided extra removal of NO_3_^−^. The additional removals are also justified based on previous studies related to OSA processes or coupling side-stream reactors to MBR and the sulfodogenic OSA processes. For instance, Zhou et al. [27] also reported the enhanced removal of TN in OSA compared to the conventional activated sludge process; similar to the current study, a slight increase in NH_4_^+^ was also found due to hydrolysis. In another study, marginally improved nutrient removal was reported due to a coupled SSR [28]. Coupling the SSR to MBR also caused a slight increase in NH_4_^+^ concentration but reduced TN in the effluent than the control MBR [10]. This study did not use any additional reactor; instead, it managed the feast/famine conditions in the M-tank and promoted in-situ sludge hydrolysis and reduction compared to all these reports.

### 3.2. Sludge Reduction

The changes in the averaged MLSS concentration during unstable and stable phases of operation are presented in Figure 2a. A relatively consistent averaged MLSS was obtained after more than 100 days of operation. The averaged MLSS concentration in the stable phase was found to be 4.40 ± 0.26, 3.55 ± 0.21, and 3.17 ± 0.23 g/L for MBR_control_, AOA_90_, and AOA_80_, respectively. The MLSS concentration in different tanks of MBR_control_, AOA_90_, and AOA_80_ was shown in Appendix A. These results presented up to a 20% and 30% reduction in MLSS with recirculation rates of 90% and 80% in AOA bioreactors, respectively.

However, the calculations of the observed yield were based on the MLSS obtained during the stable phase of operation only. The observed yield coefficient (Y_obs_) was calculated by the slope of the increase in cumulative sludge generated (g MLSS) and cumulative consumed substrate (g DOC), as presented in Figure 2b. This approach of calculating yield was adopted in several previous studies related to sludge reduction in activated sludge processes [4,28]. The factors like SRT were also considered in such calculation, which was constant for all bioreactors. The cumulative sludge generated was based on the average of MLSS in anoxic, oxic and M-tanks. The results showed that the Y_obs_ of MBR_control_ was 0.31 gMLSS/g DOC, which was relatively lower than the previous studies, with the usual value of around ~0.4 gMLSS/g COD [29,30]. Longer SRT (i.e., 80 days) could be ascribed to such lower yield in MBR_control,_ promoting endogenous respiration for cell maintenance and leading to cell lysis and decay [31,32]. Several previous studies compared the impact of different SRTs on sludge yield in MBR and other biological systems; lower yield with cryptic growth and lysis has been reported at prolonged retention conditions [33,34].

Compared to MBR_control_ (Y_obs_ = 0.31), the Y_obs_ for AOA_90_ and AOA_80_ were found to be 0.25 and 0.22 gMLSS/gDOC, respectively. These results infer that the operational scheme of AOA bioreactors successfully reduced sludge biomass up to 19.35% and 29.03% at recirculation rates of 90% and 80%, respectively. Such a reduction in biomass yield indicates the role of multiple factors, including cell lysis, hydrolysis of particulate to DOM, and cryptic growth in the M-tank of the AOA bioreactors. The gradual concentrating biomass in M-tank with lower recirculation rates was found effective in controlling the sludge yield with equivalent or better performance in removing pollutants. The higher MLSS in famine conditions instigated the phenomenon of self and in-situ reduction of biomass in the anoxic condition in M-tank.

### 3.3. Dynamics of EPS and SMP in AOA Bioreactors

The AOA configurations brought a substantial reduction in sludge growth, and such endogonesis conditions also promoted the hydrolysis of EPS, as presented in Figure 3. In MBR_control_, the total EPS in terms of average DOC/g MLSS was found in a decreased order from anoxic to oxic to M-tank; however, the difference among the units was statistically insignificant (*p* > 0.05). The AOA bioreactors showed contrasting results for corresponding units in MBR_control_ (*p* < 0.05). The AOA_90_ showed an increase in EPS concentration than MBR_control_; however, EPS in AOA_80_ was noticeably lower in each reactor than MBR_control_ and AOA_90_. The higher EPS production in the anoxic and oxic units of AOA_90_ could be due to pseudo-endogenous respiration, which probably suppressed the microbial functioning in the anoxic M-tank. Upon return to the pre-anoxic unit, microorganisms regained activity in response to feast conditions there. It is worth mentioning that F/M in both membrane-controlled AOA bioreactors shifted from the start of 24 h to the end. At 0 h, the F/M was highest in anoxic and oxic units and lowest in the M-tank and was vice-versa at the 24th hour. In AOA_80_, extreme biomass retention in the M-tank possibly caused hydrolysis of bound EPS, which inhibited the microbial functioning extensively and promoted sludge digestion. The extreme digestion seized the microbial activity for an extended period, which was even not recovered while recirculating to an anoxic chamber and followed by an oxic chamber.

The EPS contents, protein, and polysaccharides were found abundant in the anoxic and oxic units of AOA_90_ compared to AOA_80_ and MBR_control_ (*p* < 0.05) (Figure 3). The average polysaccharide vs. protein in M-tanks of MBR_control_, AOA_90_, and AOA_80_ was recorded to be 8.75 ± 3.58 vs. 14.58 ± 5.48, 8.2 ± 3.11 vs. 15.04 ± 5.74, and 7.71 ± 2.18 vs. 12.43 ± 5.64 mg/gMLSS, respectively. The results presented an insignificant difference in the average concentration of EPS in the M-tanks of all the bioreactors. It is deduced from the results that the configuration of the AOA bioreactor showed the capability to reduce biomass while keeping EPS at a matching level to MBR_control_. The changes in SMP in AOA bioreactors were also measured and presented in Appendix A. The SMP in the anoxic and oxic units in AOA_90_ and AOA_80_ could interfere with the supernatant DOC with possible low degradation; therefore, only changes in polysaccharide and protein in SMP were elucidated. The polysaccharide concentration in the M-tank of MBR_control_, AOA_90_, and AOA_80_ was 1.11 ± 0.60, 1.80 ± 1.49 and 5.46 ± 4.84 mg/L, respectively, indicating the release of intracellular polysaccharide in AOA bioreactors. The protein concentration in the M-tank of MBR_control_, AOA_90_, and AOA_80_ was 2.97 ± 2.67, 4.80 ± 2.76 and 3.18 ± 2.18 mg/L, respectively. The relatively higher protein concentration in AOA bioreactors was possibly due to cell lysis and hydrolysis that occurred during in-situ sludge reduction. The release of proteinous compounds during digestion or sludge reduction has been abundantly reported in previous studies related to anaerobic digestion and OSA processes [35]. In addition, studies have demonstrated that in anoxic conditions, SMP could be re-utilized as electron donors by bacteria for denitrification [36,37]. Considering the higher TN removal efficiency in the AOA_80_ bioreactor, it could be speculated that the generated protein was quickly metabolized by the denitrifying bacteria in the AOA_80_ bioreactor. As such, the protein concentration in the M-tank followed the order: AOA_90_ > AOA_80_ > MBR_control_.

The DOM composition in EPS and SMP could help understand the phenomenon behind the sludge reduction. Several indicators are suggested in previous studies for separating the different respiration conditions, hydrolysis, and cell lysis [38]. As such, the DOM composition and concentration changes were explored to better understand the mechanism of in-situ sludge reduction.

The maps of the three components with their peak positions of EEM-PARAFAC modeling are presented in Appendix A. One component (C1) showed a resemblance to the protein-like fluorescence, with its features matching to tryptophan-like. The other two components, C2 and C3, represent the fulvic-like and humic-like fluorescence, respectively [39,40,41,42,43,44]. The distribution of fluorescence components in the SMP and EPS from different units of bioreactors is presented in Figure 4. The fulvic-like C2 was found abundant in SMP of MBR_control_, which could be generated by the microbial degradation of organics with abundant oxygen [45]. The total fluorescence intensity was found to decrease from anoxic to oxic to M-tank, and the average reduction was recorded to around 20%. These changes could be linked to the increasing performance along the treatment train as reflected by the basic water quality parameters, such as NH_4_^+^ removal. In contrast, the protein-like component showed a higher share in SMP than fulvic-like components in AOA bioreactors. Moreover, changes in composition were noticeable among the different treatment units; the anoxic tank and M-tank showed similar composition. The SMP enriched with the protein-like component in the anoxic unit and M-tank indicated the cell lysis due to extensive biomass retention and the SMP composition was found similar (*p* > 0.05). However, a noticeable difference was recorded in total component intensity (R.U.) between the bioreactors. AOA_90_ produced a relatively lower SMP concentration as fluorescent DOM (FDOM) than AOA_80_. The average total component was higher in the anoxic tank and M-tank of AOA bioreactors than MBR_control_. Cell lysis and hydrolysis during in-situ sludge reduction could be the causes of such change in FDOM. Similarly, adding a side-stream reactor to MBR was reported to cause the higher release of FDOM, and an anaerobic sludge digester also released SMP enriched with protein [30,46].

The EPS-related FDOM was measured in terms of R.U./g MLSS and is presented in Figure 4. In all units of bioreactors, protein was noted to be the leading component with a contribution of above 85%. A direct influence of the extra sludge in terms of decreased R.U./gMLSS in M-tanks of AOA_80_ and AOA_90_ was recorded. The decreased EPS in the M-tanks of AOA bioreactors indicated the consumption of polymeric substances attached to microorganisms and inside the microbial flocs. Oxic units in AOA_90_ and AOA_80_ showed higher FDOM up to 10.0 R.U./gMLSS and 5.0 R.U./gMLSS, respectively. Among the bioreactors, AOA_80_ has the lowest R.U./gMLSS. It infers that extensive sludge reduction substantially controlled the yield of biomass and suppressed the EPS.

### 3.4. Microbial Community

The variation in sludge recirculation ratio altered the F/M as well as the microorganisms in the bioreactors. The evolution of the microbial community was monitored and presented in Appendix A and Figure 5a,b. In MBR_control_, the Simpson index first increased and then decreased, with the highest value in the mid-term operation period. In AOA_90_, an opposite trend for microbial enrichment was observed compared to MBR_control_. However, the Simpson index in AOA_80_ increased with the operation time, indicating the decreased microbial diversity. The substrate-limited environment in the M-tank caused sludge hydrolysis and thus possibly decreased the microbial diversity. Meanwhile, the richness of the microbial species decreased with time in all three bioreactors, and AOA_80_ had the lowest richness. The 80% recirculation ratio promoted the starvation conditions, which led to the biomass decay in AOA_80_. The varying recirculation rate of sludge from the membrane tank to the pre-anoxic tank would affect microbial community relationships to carry out in-situ sludge reduction in the bioreactors [10], which will be discussed in the following section.

The microbial composition was classified at the phylum level, as presented in Figure 5c. *Proteobacteria*, *Bacteroidetes*, *Firmicutes* and *Planctomycetes* were the major phyla throughout the sludge reduction process. In all the bioreactors, the relative abundance of *Proteobacteria* increased with the operation time, and AOA_80_ had the highest abundance at the 90th day of operation. *Proteobacteria* was known as the dominant hydrolytic and acidogenic bacteria with the potential for polysaccharide degradation [47]. *Bacteroidetes* played a strong role in large molecular organic matter degradation, i.e., proteins and polysaccharides [48]. The relative abundance of *Bacteroidetes* in AOA_90_ increased with the operation time and became the highest among three bioreactors. *Firmicutes* was responsible for anoxic hydrolysis and acidification [49], and was beneficial for organic degradation [50]. The relative abundance of *Firmicutes* in AOA_80_ was the highest than in other bioreactors at the 90th day of operation, in agreement with its best sludge reduction performance (Figure 2).

Further analysis of microbial community at the genus level was also conducted (Figure 5d). *Dechloromonas* and *Zoogloea* were the dominant genera of three bioreactors. *Dechloromonas*, the slow-growing bacteria with the function of denitrification [46], was the most abundant genus in AOA_80_ (34.76% at the 90th day of operation). *Zoogloea* was known as the denitrifying bacteria [23] and could facilitate the removal of organic compounds [51]. The relative abundance of *Zoogloea* in AOA_90_ was the highest among three bioreactors at the 90th day of operation. *Chryseobacterium* with the highest richness in AOA_90_ would also induce EPS degradation due to its good proteolytic ability [52]; therefore, resulting in the cell lysis in AOA_90_. These findings confirmed that the slow-growing and hydrolytic microorganisms played a critical role on the biomass reduction in AOA_90_ and AOA_80_ [53,54].

The correlation of the dominant phyla of microorganisms, sludge reduction, polysaccharide, protein and EPS was conducted via PCA analysis, as shown in Figure 5e. The respective contributions of the first and second axes to the total variance accounted for 63.37% and 36.63%, respectively. Obviously, the bacteria communities of MBR_control_, AOA_90_, and AOA_80_ bioreactors were clustered separately, which indicated that the different sludge recirculation rates induced the evolution of the microbial communities. The positive correlation between sludge reduction and *Proteobacteria*, *Firmicutes* as well as *Verrucomicrobia* could be clearly identified, especially in AOA_80_, while *Planctomycetes* had a totally negative correlation with sludge reduction in MBR_control_. Meanwhile, *Firmicutes* and *Bacteroidetes*, phylum with the capability of organic degradation [48,50], were closely positive with polysaccharide removal in AOA_90_ and protein removal in AOA_80_, respectively.

The network analysis was conducted by Gephi (v 0.9.2) [55] to uncover the possible correlation (Pearson’s correlation coefficient ρ > 0.6 and significance *p* < 0.01) between different genera in the bioreactors (Figure 5f). Different modules were distinguished by node colors, while node size was determined by the number of node connections. The orange and blue edges that connected two nodes were used to illustrate the positive and negative correlations [56]. Close positive relationships were found among dominant genera compared to negative interactions. The slow-growing bacteria *Sulfuritalea*, *Zavarinella* and *Azonexus* [26,54] exhibited a positive correlation with many other genera. *Sulfuritalea* was positively correlated with *Lacibacter*, *Pirellula* and *Reyranella*, which were able to hydrolyze organic compounds [57,58,59]. *Zavarinella* also showed a positive relationship with *Pirellula* and *Reyranella.* However, there was a negative correlation between *Roseomonas* and *Sulfuritalea*, which might be due to the competition or niche exclusion between these two species under limited substrate conditions [56]. *Azonexus*, *Comamons* and *Vampirovibrio* belonged to the same module, and *Azonexus* exhibited a positive relationship with *Comamons* and *Vampirovibrio*, the hydrolytic bacteria [60] and predatory bacteria [61], respectively. Moreover, the fermentative bacteria *Tolumonas* [62] correlated positively with *Anaerolinea*, one of the slow glowing species [63]. Herein, positive networks were found between the slow-growing species, hydrolysis and fermentative bacteria, contributing to sludge reduction [56].

### 3.5. Metabolic Response Pathways

In spite of the change in the microbial community structure in three bioreactors, different sludge recirculation rates would also regulate the expression of functional genes. In this section, the carbon and nitrogen metabolic characteristics of three bioreactors were expounded.

#### 3.5.1. Carbohydrate Metabolic Analysis and Functional Genes

Glycolysis is a significant part of carbohydrate metabolism, which converts glucose into pyruvate with energy liberation [48]. As presented in Figure 6, glucose was first converted into glucose-6P under the function of glucokinase (EC: 2.7.1.2) and polyphosphate glucokinase (EC: 2.7.1.63). The average relative abundance of glucokinase (polyphosphate glucokinase) was 0.0149% (0.0032%), 0.0165% (0.0023%), and 0.0186% (0.0027%) in MBR_control_, AOA_90_ and AOA_80_, respectively, indicating that AOA_80_ exhibited a higher proportion of key genes encoding enzymes in this step. Then, glucose-6-phosphate isomerase (EC: 5.3.1.9) catalyzed glucose-6P into fructose-6P and fructose-1, while 6P2 was formed with the function of 6-phosphofructokinase (EC: 2.7.1.11) followed by the triose phosphate isomerase reaction [48]. Later on, glyceraldehyde-3P was transformed into phosphoenolpyruvate after a series of reactions with the conversion between NAD^+^ and NADH along with ATP generation. Finally, pyruvate was produced with the catalysis of pyruvate kinase (EC: 2.7.1.40) and could be further metabolized into CO_2_ [64]. The total relative abundance of genes encoding enzymes involved in the whole glycolysis process was 0.171%, 0.172%, and 0.184% in MBR_control_, AOA_90_ and AOA_80_, respectively, suggesting that more energy would be provided in AOA_80_ for microbial metabolism and organic degradation. During the glucose degradation process, the fructose-1, 6P2 and pyruvate production were the most important reactions, among which the former step was the rate-limiting procedure that controlled the whole glycolysis process [64]. The average proportions of genes encoding the two enzymes (EC: 2.7.1.11 vs. EC: 2.7.1.40) for the production of fructose-1, 6P2 and pyruvate in MBR_control_, AOA_90_ and AOA_80_ were 0.0217% vs. 0.02%, 0.0244% vs. 0.022% and 0.0232% vs. 0.023%, respectively. This indicated that the glycolysis rate would be faster in AOA_90_ and AOA_80_ compared to MBR_control_, implying enhanced microbial activity in AOA_90_ and AOA_80_ [65]. As such, the utilization rate of secondary substrates would be promoted in AOA_90_ and AOA_80_, which possibly resulted in improved sludge reduction efficiency.

#### 3.5.2. Nitrogen Metabolic Analysis and Functional Genes

The nitrogen metabolic pathway and the comparative analysis of functional genes in three bioreactors were investigated as shown in Figure 7. The total relative abundance of genes encoding enzymes for the nitrification process was highest in MBR_control_, followed by AOA_90_ and AOA_80_. The proportion of genes encoding the hydroxylamine dehydrogenase (EC: 1.7.2.6) reduced by 22.95% (AOA_90_) and 35.86% (AOA_80_) compared to MBR_control_. Meanwhile, the contents of the genes encoding enzymes for ammoniation (EC: 1.14.99.39) and nitrite oxidoreductase (EC: 1.7.99.-) were highest in MBR_control_. This was in good agreement with the lower concentration of NH_4_^+^ in MBR_control_ as shown in Appendix A. In contrast, the total relative abundance of genes encoding enzymes for the denitrification process was highest in AOA_80_, followed by AOA_90_ and MBR_control_. The genes encoding the enzyme responsible for the reduction of nitrate to nitrite (EC: 1.9.6.1) and the genes encoding enzymes responsible for the reduction of nitrite and nitric oxide (EC: 1.7.2.1 and EC: 1.7.2.5) followed the order: AOA_80_ > AOA_90_ > MBR_control_. Meanwhile, the average relative abundance of nitrous-oxide reductase (EC: 1.7.2.4) in AOA_90_ was the highest (0.083%), followed by MBR_control_ and AOA_80_ (0.071%). In addition, compared to MBR_control_, the abundance of key genes related to denitrification, i.e., napAB, norBC and nosZ in AOA_90_ and AOA_80_ at the 90th day of operation increased by 44.88% vs. 139.30%, 33.08% vs. 162.77%, and 50.83% vs. 12.19%, respectively (Appendix A), indicating the enhanced denitrification process in AOA bioreactors. This was indeed the case and was in agreement with the higher removal efficiency of TN in AOA bioreactors as shown in Appendix A. The organics released from cell lysis and POM hydrolysis could be utilized as carbon sources for denitrification [66]. As such, more substances could be degraded in AOA bioreactors (especially in AOA_80_), and thus resulting in the improved sludge reduction efficiency [67].

## 4. Conclusions

The AOA bioreactors efficiently reduced sludge generation through promoting hydrolysis and endogenous respiration in the anoxic M-tank. Up to 30% sludge reduction with improved denitrification was achieved in the AOA_80_ bioreactor. The decreased EPS in the anoxic M-tank of the AOA_80_ bioreactor indicated the consumption of polymeric substances attached to microorganisms and inside the microbial flocs. The higher abundance of *Dechloromonas* and *Zooglea* in AOA bioreactors facilitated the sludge reduction. In addition, metagenomics analysis uncovered that AOA bioreactors exhibited higher proportions of genes encoding enzymes involved in the glycolysis and denitrification processes, which contributed to the utilization of carbon sources as well as nitrogen consumption and thus sludge reduction. In spite of the economic benefits, this approach would help in lessening the waste sludge-related environmental issues.

## Figures and Tables

**Figure 1 membranes-12-00659-f001:**
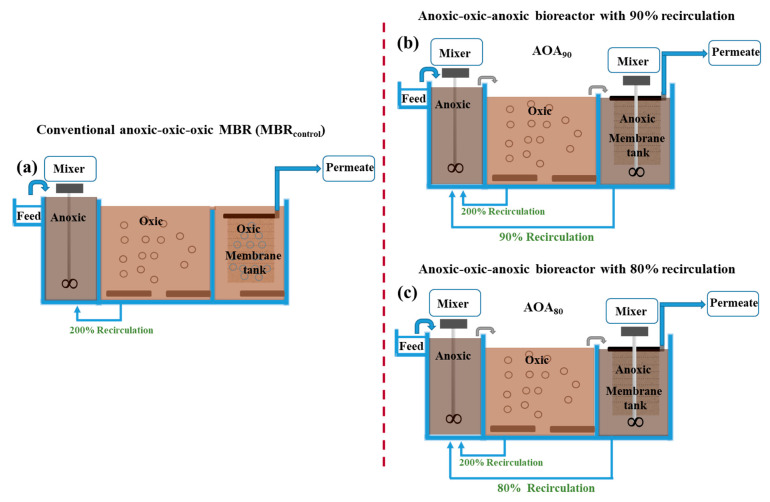
Configuration of conventional anoxic-oxic-oxic MBR (MBR_control_) (**a**) and membrane controlled anoxic-oxic-anoxic bioreactors with 90% (AOA_90_) (**b**) and 80% (AOA_80_) (**c**) recirculation.

**Figure 2 membranes-12-00659-f002:**
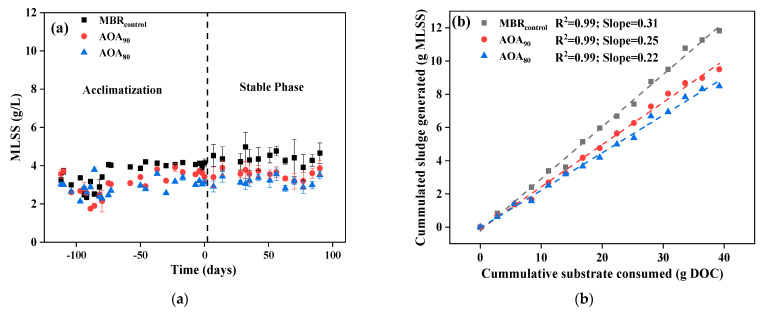
(**a**) The change in the averaged MLSS concentration between acclimatization and stable operation phases in MBR_control_, AOA_90_ and AOA_80_, (**b**) The observed sludge yield (Y_obs_) in MBR_control_, AOA_90_ and AOA_80_.

**Figure 3 membranes-12-00659-f003:**
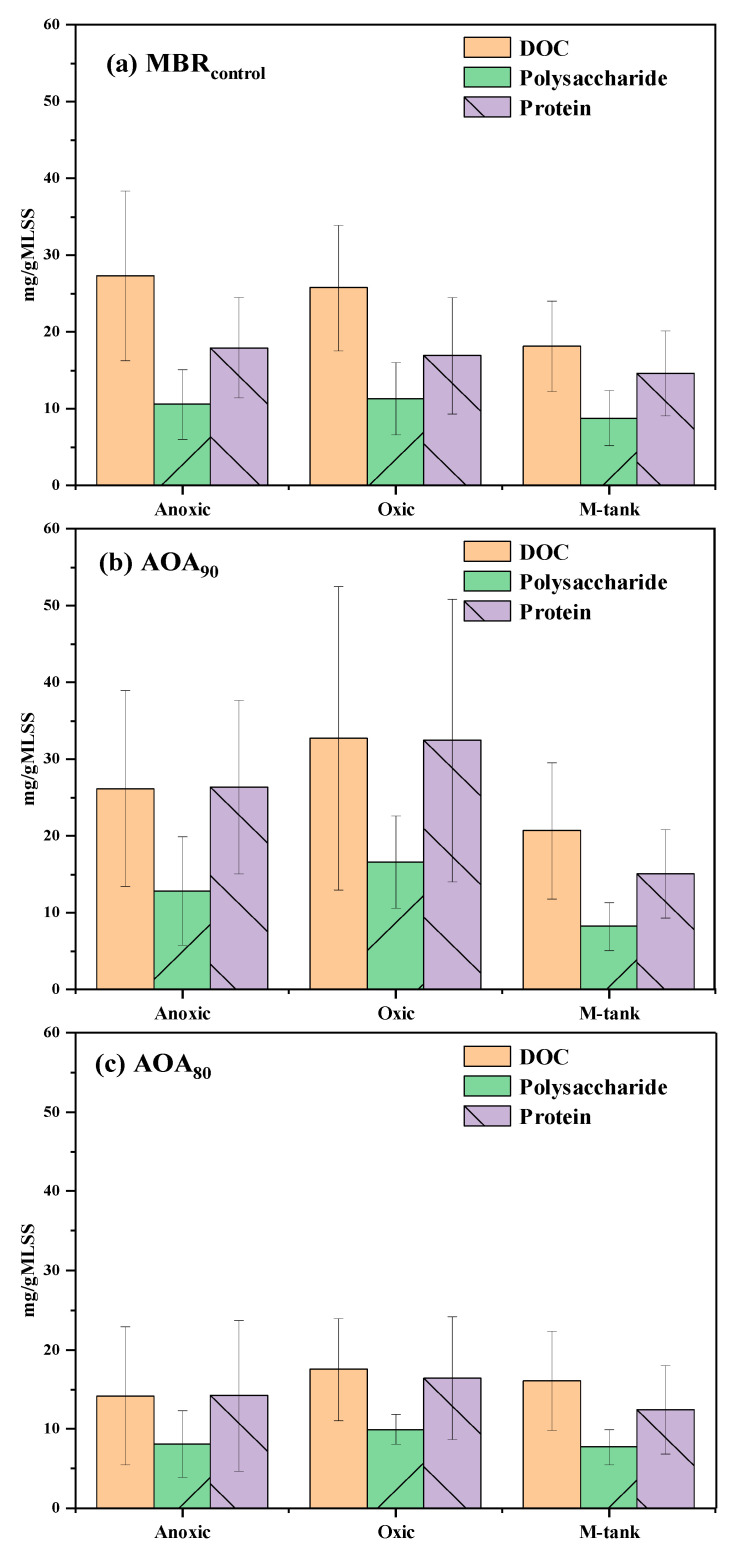
The average of total EPS in terms of DOC, polysaccharides, and protein in different units of bioreactors (*n* = 14).

**Figure 4 membranes-12-00659-f004:**
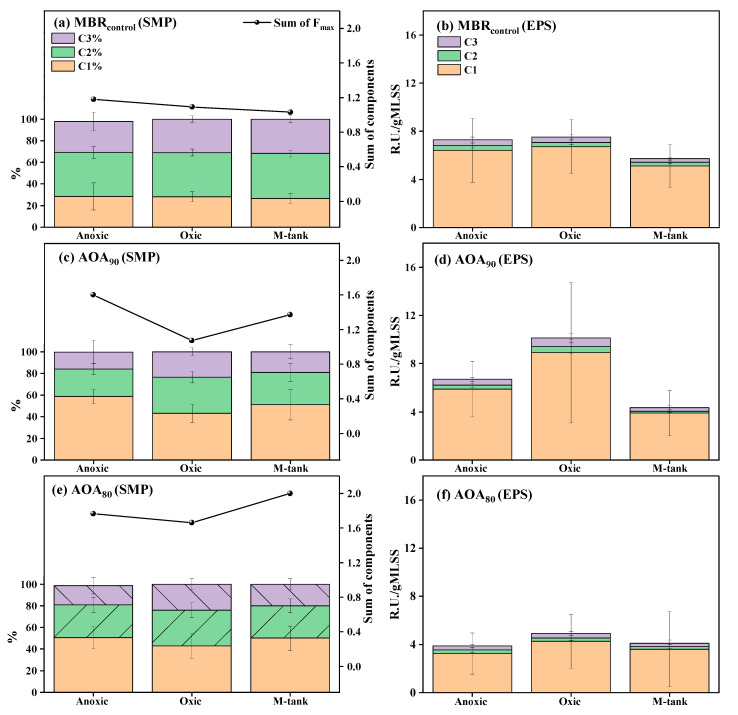
The average distribution of EEM-PARAFAC components and sum of F_max_ (C1 + C2 + C3) in SMP and EPS of different units in (**a**,**b**) MBR_control_, (**c**,**d**) AOA_90_, and (**e**,**f**) AOA_80_ (*n* = 14).

**Figure 5 membranes-12-00659-f005:**
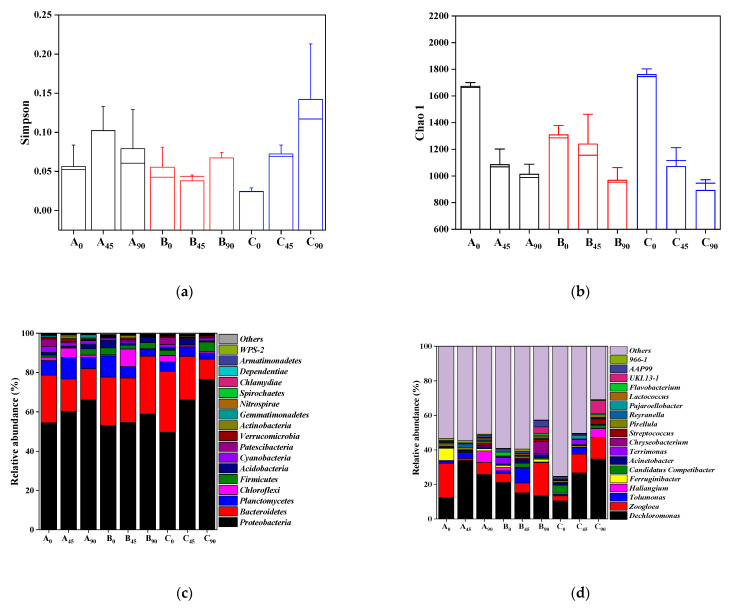
The Simpson index (**a**), Chao 1 index (**b**), relative abundance of dominant species at the phylum level (**c**) and genus level (**d**), PCA between bacteria community at the phylum level and sludge reduction (**e**) and co-occurrence network (**f**). (A, B, and C represent MBR_control_, AOA_90_ and AOA_80_, respectively, and the subscript of A, B, and C means the sampling time).

**Figure 6 membranes-12-00659-f006:**
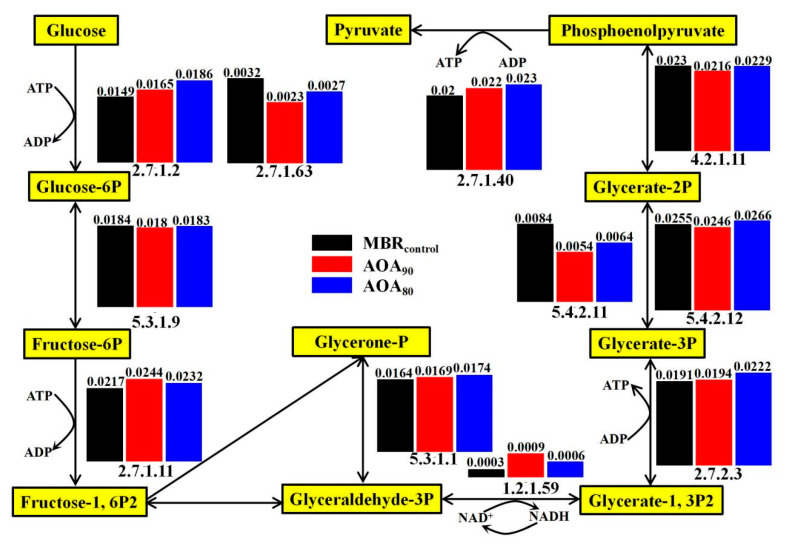
Diagram of glycolysis pathway and the average relative abundance of genes encoding enzymes (identified by their EC number) in MBR_control_, AOA_90_ and AOA_80_.

**Figure 7 membranes-12-00659-f007:**
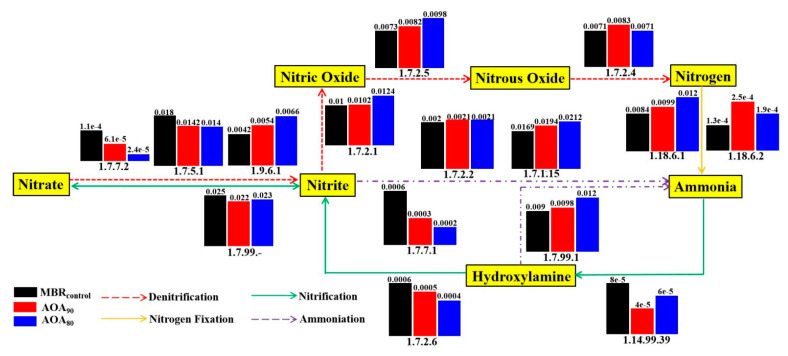
Diagram of nitrogen pathway and the average relative abundance of genes encoding enzymes (identified by their EC number) in MBR_control_, AOA_90_ and AOA_80_.

## Data Availability

All data are available in the main text or Appendix A.

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
