# Peer review of "In-Situ Sludge Reduction in Membrane-Controlled Anoxic-Oxic-Anoxic Bioreactor: Performance and Mechanism"

_membranes, 2022, doi:10.3390/membranes12070659_

Round 1

Reviewer 1 Report

In my opinion the manuscript is original and gives interesting results about a well structured and organized research.

Reviewer 2 Report

Review of the paper “In-situ sludge reduction in membrane-controlled anoxic-oxic-anoxic bioreactor: Performance and mechanism” by Chengyue Li, Tahir Maqbool, Hongyu Kang and Zhenghua Zhang.

This paper presents the experimental results of the performance of membrane-controlled anoxic-oxic-anoxic (AOA) bioreactors for in-situ sludge reduction compared to the conventional anoxic-oxic-oxic membrane bioreactor (MBRcontrol). The idea of this paper is very significant from the socio-economic and environmental aspects. However, there are several comments that should be effectively addressed.

Remarks:

1. Page 2, Line 77, Provide the full name of the term “DOM”.

2. Page 3, Line 121, The abbreviation “SMP” should be in the brackets.

3. Page 5, Line 191, Figure S2 (Supplementary Information). Please provide the uniform scale for all graphs in Figure S2 (the same min and max values for graphs a), c) and e); The same min and max values for graphs b), d) and f)) to enable easier following of the adequate explanation in the text.

4. Page 5, Line 217, Figure S3 (Supplementary Information). Please provide the uniform scale for all graphs in Figure S3. Furthermore, why are the MLSS values in M-tank significantly higher for AOA bioreactors compared to MBRcontrol? Please, explain.

5. Page 7, Line 276, Figure S4 (Supplementary Information). Please, provide the uniform scale for all graphs in Figure S4.

6. Page 7, Line 282, In the sentence “The relatively higher protein concentration in AOA bioreactors…” the conclusion is generalized for both AOA bioreactors, although the values of the protein concentration in the M-tank of MBRcontrol and AOA80 are close, while the value of the protein concentration in the M-tank of AOA90 is higher than the mentioned ones. Please, explain.

7. Page 8, Line 298, C2 represents fulvic-like fluorescence, not humic-like.

8. Page 9, Line 301, “fulvic-like C2”, not “humic-like C2”.

9. Page 10, Line 306, “humic-like” or “fulvic-like”?

10. Page 12, Figure 5 a) and b), Simpson index and Chao 1 index for B0 and C0 are not in accordance with Table S4.

11. Page 13, Lines 391-398, In the PCA diagram analysis, you have not mentioned anything about bioreactors, although the bioreactors figure in the PCA diagram.

12. The significant improvement of the conclusion is required.
